

# ETCS
## Uproszczony System Kolejowy ETCS z aplikacją Unity

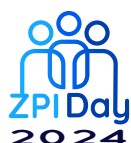

**Autorzy**: Alex Jawny ⃝ · Bartosz Pyzio ⃝ · Jakub Szwiec ⃝ · Marcin Tkocz ⃝ · Rafał Trzak ⃝

**Opiekun:** Michał Kędziora

## 1   WPROWADZENIE

Kolejowy system bezpieczeństwa ETCS jest nieustandaryzowany, a jego kod źródłowy jest niedostępny publicznie. Skutkuje to nie tylko wyższymi kosztami utrzymania systemu, ale także niższymi prędkościami kolei oraz mniejszym bezpieczeństwem podróży. Celem projektu było dostarczenie środowiska testowego i prezentacyjnego uproszczonego systemu ETCS, które ma być pierwszym krokiem w procesie ujednolicania i standaryzacji tego systemu. Aplikacja maszynisty i serwer ETCS są implementacjami rzeczywistych komponentów systemu, które w przyszłości mogą być udoskonalane w ramach wykonanego środowiska testowego. Zaprojektowana aplikacja Unity symuluje rzeczywiste środowisko i ruch pociągu, z jednej strony stanowiąc źródło danych dla pozostałych komponentów, z drugiej strony będąc wizualizacją decyzji podejmowanych przez system.

## 2   ROZWINIĘCIE

### 2.1   Wstęp

System ETCS (ang. European Train Control System) jest systemem sterowania ruchem pociągu, którego głównym zadaniem jest zwiększenie bezpieczeństwa podróży oraz prędkości pociągów. Mimo swojej ważnej roli system nie jest w pełni ustandaryzowany – w różnych krajach implementuje się w różny sposób, zarówno pod względem zakresu funkcjonalności, jak i charakterystyki procesów i dostępnych parametrów. Z kolei kody źródłowe poszczególnych implementacji nie są dostępne do wglądu, co znacznie ogranicza możliwość wykrywania luk bezpieczeństwa i nieefektywnych fragmentów kodu, a także utrudnia wspomnianą już standaryzację. Rozwiązaniem powyższych problemów jest rozpoczęcie prac na ustandaryzowaną wersją systemu ETCS oraz zapewnienie środowiska, w którym jego komponenty będą mogły być stale udoskonalane. Rozwój ustandaryzowanego systemu zapewni

- spadek kosztów związanych z utrzymaniem i rozwojem.

- zwiększone bezpieczeństwo podróży kolejowych.

- większe prędkości pociągów, które zwiększą zarówno satysfakcję podróżujących, jak i zyski generowane przez przejazdy pociągów towarowych.

Z kolei środowisko testowe i prezentacyjne przyczyni się do stymulacji innowacji w zakresie bezpieczeństwa kolei. Zainteresowane strony takie jak przedsiębiorstwa kolejowe, uniwersytety, naukowcy i studenci, będą miały szansę na zapoznanie się z działaniem obecnej wersji systemu oraz samodzielnym projektowaniem i testowaniem nowych rozwiązań. W efekcie zwiększy się efektywność implementowanych rozwiązań, a ewentualne podatności będą szybko wykrywane.

System ETCS składa się z dwóch podstawowych składowych: aplikacji maszynisty, działającej w lokomotywach pociągów, oraz serwera ETCS, który dostarcza aplikacji maszynisty informacje niezbędne do podejmowania decyzji. Dodatkowo konieczne jest zaimplementowanie komponentu symulującego rzeczywiste środowisko – tory kolejowe i rozjazdy, semafory, przejazdy kolejowe oraz poruszające się pociągi. Wobec powyższych, w ramach projektu postanowiono zrealizować następujące cele:

- Implementacja serwera ETCS – głównym zadaniem serwera jest przechowywanie informacji o torowisku, rejestrowanie zmian na trasie (zmiany semaforów, rozjazdów, przejazdów kolejowych) oraz wydawanie pociągom Zezwolenia Na Jazdę na konkretne odcinki trasy. Zezwolenia muszą być wydawane na żądanie pociągu oraz automatycznie, w wypadku zmiany stanu torowiska. Komunikacja z aplikacją maszynisty musi być ponadto szyfrowana.

- Implementacja aplikacji maszynisty – na podstawie informacji otrzymanych z serwera, aplikacja maszynisty oblicza zalecane prędkości na poszczególnych odcinkach trasy i w zależności od wyniku obliczeń informuje maszynistę o konieczności zwolnienia lub automatycznie inicjuje hamowanie pociągu. Aplikacja może działać w różnych trybach o różnych funkcjonalnościach. Do jej zadań należy także szyfrowanie i deszyfrowanie komunikatów oraz walidacja wiadomości odbieranych z serwera.

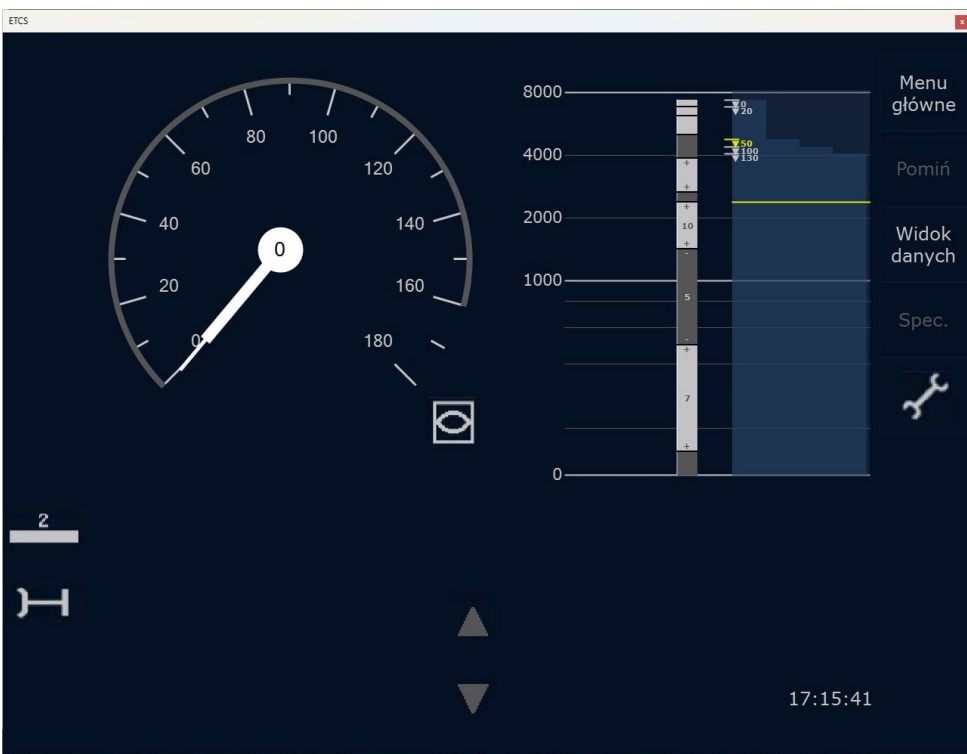

- Implementacja aplikacji Unity – jej głównym zadaniem jest zapewnienie wizualizacji ruchu pociągu. Podczas ruchu pociągu, w wyniku mijania balis zamocowanych na torach, aplikacja Unity informuje aplikację maszynisty o aktualnej pozycji pociągu. Oprócz tego, do zadań aplikacji należy zapewnienie możliwości zmiany prędkości pociągu oraz zmiany stanu elementów torowiska – rozjazdów, semaforów oraz przejazdów kolejowych.

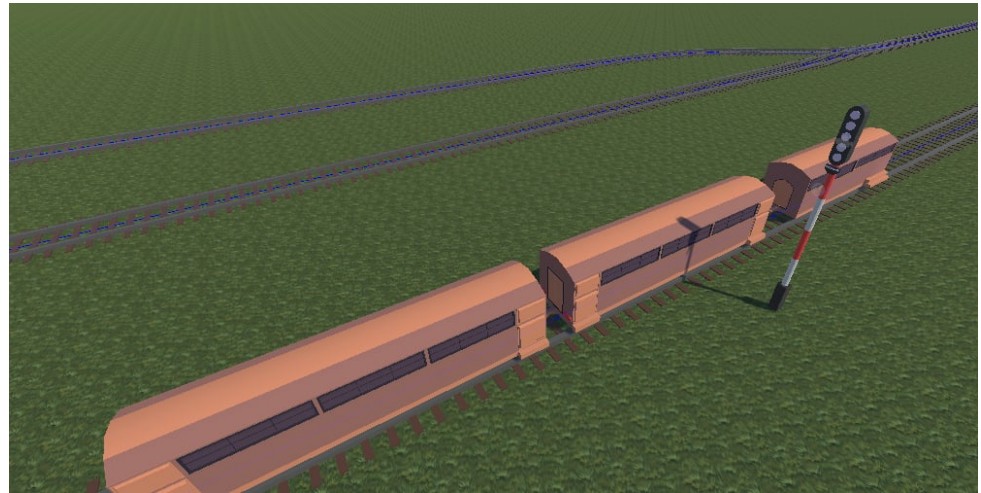

## 2.2   Prace powiązane

Jednym z ważniejszych istniejących projektów dotyczącym systemu ETCS jest openETCS. Celami tego projektu było stworzenie zestawu narzędzi do modelowania, walidacji i testowania, przeznaczonych do tworzenia wydajnych implementacji systemu ETCS. Podjęte zostały między innymi problemy formalnej specyfikacji i weryfikacji wymagań systemowych oraz generowanie i wykonywanie przypadków testowych. Istnieją także różnego rodzaju symulatory ruchu pociągu, takie jak Train Simulation Classic 2024, jednak większość z nich nastawionych jest na rozrywkę, a nie edukacje i standaryzację systemu ETCS. Pojedyncze propozycje systemu ETCS typu open source są okrojone pod względem komponentów i funkcjonalności, brakuje im także środowiska testowego i prezentacyjnego. Nasz projekt ma wypełnić tę lukę, zapewniając zarówno otwarte implementacje komponentów systemu ETCS z ich kluczowymi funkcjonalnościami, jak również symulację środowiska i ruchu pociągu.

Założono, że aplikacja symulująca rzeczywiste środowisko zostanie wykonana w Unity z wykorzystaniem języka C#. Komponenty systemu ETCS miały zostać zaimplementowane w tym samym języku – aplikacja maszynisty w środowisku .NET Framework, natomiast serwer ETCS – w ASP.NET Core. Aplikacja maszynisty miała zostać zaimplementowana jako aplikacja desktopowa, gdyż w takiej formie występuje w rzeczywistości. Do rozwoju projektu wybrane zostało środowisko Visual Studio, oferujące wsparcie zarówno dla samego języka C#, jak i Unity, .NET oraz ASP.NET Core. Do zarządzania wersjami kodu wykorzystano platformę GitHub, natomiast do planowania i harmonogramowania zadań wykorzystano Jirę. Prace prowadzono w filozofii Agile, ze sprintami o długości od jednego do dwóch tygodni.

Ze względu na krótki czas implementacji projektu niemożliwe było wdrożenie całego systemu ETCS ze wszystkimi jego funkcjonalnościami. Przykładowo, w ramach aplikacji maszynisty zaimplementowano najważniejsze tryby działania, pomijając te o mniejszym znaczeniu. Innym ograniczeniem była niepełna i niespójna dokumentacja związana z brakiem standaryzacji – mimo wzorowania naszego projektu na systemie ETCS, pewne decyzje musiały zostać podjęte arbitralnie w celu zamodelowania procesów zachodzących w systemie.

## 2.3   Rezultaty

Projekt udało się zrealizować w zamierzonym czasie i w zakresie założonych funkcjonalności. W wyniku działań całego zespołu, powstały trzy kluczowe komponenty: aplikacja maszynisty, serwer ETCS oraz środowisko testowo-prezentacyjne Unity.

Aplikacja maszynisty jest wzorowana na rzeczywistym interfejsie spotykanym w lokomotywach pociągów. Było to możliwe dzięki materiałom dostępnym na platformie YouTube, m.in. dzięki Krzysztofowi Waszkiewiczowi oraz kanałom Pitkowa Kolej i 85okN. Przed rozpoczęciem jazdy maszynista ma możliwość wprowadzenia informacji o pociągu, m.in. długości i prędkości maksymalnej pociągu, które są następnie przesyłane do serwera. Aplikacja nasłuchuje na informacje z balis wysyłane przez aplikację Unity, na podstawie których określa swoją aktualną pozycję i prędkość. Te informacje także są przesyłane do serwera i stanowią podstawę udzielania Pozwolenia Na Jazdę. Gdy aplikacja maszynisty otrzyma takie pozwolenie, jest ono przetwarzane w celu podjęcia decyzji o maksymalnej dozwolonej prędkości. Na podstawie swojego aktualnego położenia, aplikacja oblicza krzywe hamowania – jest to informacja o miejscu, w którym należy rozpocząć hamowanie, w zależności od aktualnej prędkości pociągu. Taka informacja jest wyświetlana na interfejsie graficznym w celu poinformowania maszynisty o zbliżającej się konieczności zmniejszenia prędkości. Gdy pociąg jedzie zbyt szybko i nie ma możliwości wyhamowania przed określonym końcem Pozwolenia Na Jazdę, aplikacja automatycznie inicjuje hamowanie. Oprócz tego, aplikacja wyświetla aktualną prędkość pociągu, gradienty nadchodzących odcinków torów, stan połączenia z serwerem czy wiadomości tekstowe odbierane przez aplikację. W celu poprawnego funkcjonowania systemu, aplikacja rejestruje pociąg na serwerze w momencie wjeżdżania w obszar ETCS i wyrejestrowuje go, gdy wyjeżdża z tego obszaru. Obsługiwana jest także automatyczna zmiana trybu działania aplikacji, gdy odpowiednia balisa poinformuje o możliwości przejścia do trybu ETCS.

Serwer ETCS jest komponentem wspierającym aplikację maszynisty przez zarządzanie Pozwoleniami Na Jazdę. Jest to możliwe przede wszystkim dzięki przechowywaniu informacji o torowisku – rozłożeniu poszczególnych linii i torów, rozjazdów i ich możliwych konfiguracji, przejazdów kolejowych, semaforów i wskaźników. Te informacje serwer przechowuje w bazie danych, dzięki czemu możliwa jest realizacja głównego zadania serwera – wydawanie poprawnego Zezwolenia Na Jazdę. Przed wydaniem zezwolenia serwer sprawdza, czy pociąg może je otrzymać – walidacja obejmuje między innymi rejestrację pociągu,

jego pozycję i kierunek jazdy. Jeśli pociąg spełnia warunki do otrzymania Pozwolenia Na Jazdę, serwer formuje odpowiedni komunikat, uwzględniając w nim dozwolone prędkości na kolejnych odcinkach, gradienty torowiska, informacje o zmianach linii oraz ewentualne komunikaty tekstowe dla maszynisty. Poprawność działania tego mechanizmu została sprawdzona z wykorzystaniem opracowanego zestawu scenariuszy – dla utworzonego wycinka torowiska oraz pozycji pociągu określano oczekiwane Zezwolenie Na Jazdę, a następnie sprawdzano, czy dla danych warunków serwer zwracał oczekiwane wartości. Drugą odpowiedzialnością serwera jest reagowanie na zmiany w świecie Unity przed odbieranie wiadomości z odpowiednich endpointów. W wypadku zmiany stanu semafora, rozjazdu lub przejazdu kolejowego, serwer identyfikuje pociągi, do których należy wysłać nowe Zezwolenie Na Jazdę, a następnie formuje nowe zezwolenia i wysyła je do aplikacji maszynisty. Także te funkcjonalności zostały pokryte odpowiednimi scenariuszami testowymi.

Aplikacja Unity zapewnia źródło danych dla aplikacji maszynisty oraz wizualizację podejmowanych przez nią decyzji. W ramach zaprojektowanego wycinka rzeczywistości, w aplikacji stworzono torowisko o długości niemal 15 kilometrów, z rozjazdami, przejazdem kolejowym, semaforami świetlnymi oraz balisami. Gdy pociąg w aplikacji najeżdża na zamocowaną przy torach balisę, aplikacja maszynisty jest informowana o nowej lokalizacji pociągu. Interfejs umożliwia sterowanie prędkością pociągu, jak również zmianę stanu elementów torowiska – semaforów, rozjazdów i przejazdu kolejowego. Dzięki temu możliwe jest testowanie zachowania systemu poprzez sprawdzenie reakcji na zmiany torowiska. Aplikacja spełniła więc założone zadania, umożliwiając testy i wizualizację zaimplementowanego systemu ETCS.

Po implementacji, komponenty systemu ETCS zostały szczegółowo przetestowane. Serwer ETCS został pokryty testami jednostkowymi i systemowymi, sprawdzającymi wyjścia systemu dla poszczególnych wejść i stanów wewnętrznych. Stworzono kilkanaście scenariuszy, dla których najpierw sformułowano oczekiwane wyniki, a następnie porównano je z wyjściami serwera. Wszystkie testy dały oczekiwane rezultaty. Przez ostatnie dwa tygodnie projektu przeprowadzano także testy manualne, aby upewnić się, że komunikacja między komponentami przebiega w zamierzony sposób.

# 3 ZAKOŃCZENIE

## 3.1 Wnioski

W wyniku prac udało się stworzyć kompletne środowisko testowe i prezentacyjne do uproszczonej wersji systemu ETCS, dostępnej na zasadach open source. Wierzymy, że jest to ważny krok w stronę popularyzacji i standaryzacji systemu, co przyczyni się do tańszych i szybszych podróży kolejowych. Standaryzacja ułatwi współpracę krajową i międzynarodową, wspierając rozwój innowacji i dalsze doskonalenie systemu. Otwarty kod i przeglądy ekspertów umożliwią wczesne wykrywanie luk bezpieczeństwa oraz identyfikację nieefektywnych procesów. System ETCS ma w sobie ogromny potencjał, a nasz system przyczyni się do jego rozwoju.

## 3.2 Przyszłe kierunki rozwoju

Jesteśmy świadomi, że projekt jest początkiem prac związanych z doskonaleniem i standaryzacją systemu ETCS. Z powodu krótkiego czasu projektu nie było szansy zaimplementować wszystkich funkcjonalności systemu. W przyszłości do aplikacji maszynisty planujemy dodać brakujące tryby, między innymi System Failure, występujący w przypadku awarii systemu. Z kolei serwer może zostać udoskonalony przez dodanie brakujących poleceń i komunikatów, na przykład Polecenia Awaryjnego Zatrzymania Pociągu, wysyłanego w sytuacjach awaryjnych. Dalszego rozwoju można też spodziewać się w wypadku aplikacji Unity, dla której przykładowe udoskonalenia obejmują dodanie większej liczby pociągów oraz możliwość modyfikacji mapy torowiska. Obecna wersja systemu stanowi więc znakomity punkt wyjściowy do rozwoju nowego, ustandaryzowanego i opartego na otwartym kodzie źródłowym systemu ETCS, który zrewolucjonizuje transport kolejowy.

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
