# OpenReview forum: "Uproszczony System Kolejowy ETCS z aplikacją Unity"
_pwr.edu.pl/Wrocław_University_of_Science_and_Technology/2024/ZPI_Day — Wrocław University of Science and Technology 2024 ZPI Day Submission_

### Official Review · Reviewer_7qvQ · 2024-12-03
**Mediocre implementation of an important system**

**Confidence:** 4
**Significance Of Results:** 4
**Overall Quality:** 3

**Compliance With Template:**

3: Average Quality – The article includes most of the required sections, but some may be incomplete, written in a general or unclear manner. The content is correct but requires further refinement.

**Description Of Results:**

3: Average Quality – The results are described with moderate detail. Some examples or evaluation elements are present but insufficiently developed or incomplete.

**Feedback On Consistency:**

The description is consistent, but lacking in details on the implementation and applicability. On one hand it seems the authors assume some train knowledge from readers, on the other - it is hard to gauge how close it is to a computer game, a simulator, or an app to compliment the train staff. The flow of text is logical, but the testing and proper conclusions are missing.

**Potential For Development:**

The authors seem to think that they can deploy it, but I don't think the technology used is proper for profesional environement. But it can be redesigned as a simulation tool.

**Project Nature Evaluation:**

As a basic engineering work it is OK, but is lacking in any form of testing. It is hard to say if the used technical approaches are correct to final deployment - if it is a game or simulation then yes, but if it were to complement real train soft- and hardware then probably it is not robust enough - that is what testing is for.

**Technical Language Precision:**

3: Average Quality – The language is mostly appropriate but may contain minor terminological or stylistic errors. Some statements might lack precision or require improvement for better readability.

---

### Official Review · Reviewer_9buV · 2024-12-03
**Zamierzenie ambitne jednak prezentacja rezultatów dość spłycona.**

**Confidence:** 4
**Significance Of Results:** 2
**Overall Quality:** 3

**Compliance With Template:**

3: Average Quality – The article includes most of the required sections, but some may be incomplete, written in a general or unclear manner. The content is correct but requires further refinement.

**Description Of Results:**

3: Average Quality – The results are described with moderate detail. Some examples or evaluation elements are present but insufficiently developed or incomplete.

**Feedback On Consistency:**

>>> 1. Język:

Język raportu jest poprawny a myśli wyrażone są w sposób zrozumiały. Pojawiają się jedynie nieliczne błędy, np. niepoprawne użycie znaków diakrytycznych w wypunktowaniu:

"Rozwój ustandaryzowanego systemu zapewni
• spadek kosztów związanych z utrzymaniem i rozwojem.
• zwiększone bezpieczeństwo podróży kolejowych.
• większe prędkości pociągów, które zwiększą zarówno satysfakcję podróżujących, jak i zyski generowane przez przejazdy pociągów towarowych."
=>
"Rozwój ustandaryzowanego systemu zapewni:
• spadek kosztów związanych z utrzymaniem i rozwojem,
• zwiększone bezpieczeństwo podróży kolejowych,
• większe prędkości pociągów, które zwiększą zarówno satysfakcję podróżujących, jak i zyski generowane przez przejazdy pociągów towarowych."

>>> 2. Prezentacja:

2.1 Brak odniesień do literatury w tekście.
2.2 Ilustracje nie są numerowane, nie mają tytułów
2.3 Brak sekcji "Abstrakt"
3.4 Brak graficznej prezentacji struktury systemu utrudnia zrozumienie co zostało wykonane.
3.5 Nie zostały zaprezentowane wyniki symulacji - czy można było oszacować, np. jak wytworzone oprogramowanie (aplikacja maszynisty obliczająca zalecane prędkości) zwiększa bezpieczeństwo podróżnych?

>>> 3. Inne problemy:

3.1 Rozmieszczenie treści w poszczególnych sekcjach tekstu nie zawsze jest trafne. Np. punkt "2.2 Prace powiązane" prócz tego na co tytuł wskazuje, zawiera też opis wyboru technologii realizowanego rozwiązania oraz jego ograniczenia.
3.2 Tekst zawiera stwierdzenia niepoparte przedstawionym materiałem, np.:
- "a nasz system przyczyni się do jego rozwoju." (Na pewno?)
- "Obecna wersja systemu (..) zrewolucjonizuje transport kolejowy." (Na pewno? W jakim aspekcie?)

===EOT===

**Potential For Development:**

Yes.

**Project Nature Evaluation:**

Yes, the project exhibits characteristics of an engineering work, with high level of utility, application of technical methods, and technological solutions.

**Technical Language Precision:**

3: Average Quality – The language is mostly appropriate but may contain minor terminological or stylistic errors. Some statements might lack precision or require improvement for better readability.

---

### Official Review · Reviewer_W3Vc · 2024-12-03
**Uproszczony System Kolejowy ETCS z aplikacją Unity**

**Confidence:** 4
**Significance Of Results:** 5
**Overall Quality:** 4

**Compliance With Template:**

5: Very High Quality – The article contains all the required sections, which are written in a very detailed, clear, and error-free manner. The structure is professional and meets expectations, and the content adheres to the highest substantive and formal standards.

**Description Of Results:**

5: Very High Quality – The results are described in detail, clearly and comprehensively, supported by thorough evaluation, analysis, and convincing usage examples. The description meets the highest substantive standards.

**Feedback On Consistency:**

Consistency of the project description is High Quality. The problem analysis, presentation of results, and conclusions are consistent and logical.

**Potential For Development:**

Current project status is good entry point for future development of upgraded solution with new functionality.

**Project Nature Evaluation:**

Project exhibit characteristics of an engineering work, such as the level of utility, application of technical methods, and technological solutions. In general project implements ETCS system simulator.

**Technical Language Precision:**

5: Very High Quality – The language is entirely appropriate for a technical report. All terms are used correctly and precisely, and the style is professional, clear, and coherent, without any errors or ambiguities.

---

### Decision · Program_Chairs · 2024-12-10

Accept (Poster)